# Effects of Different Physical Activity Levels during a Single Day on Energy Intake, Appetite, and Energy Balance: A Preliminary Study

**DOI:** 10.3390/nu11030690

**Published:** 2019-03-23

**Authors:** Yoichi Hatamoto, Rie Takae, Ryoma Goya, Eiichi Yoshimura, Yasuki Higaki, Hiroaki Tanaka

**Affiliations:** 1Institute for Physical Activity, Fukuoka University, 8-19-1 Nanakuma, Jonan-ku, Fukuoka 814-0180, Japan; higaki@fukuoka-u.ac.jp (Y.H.); monkeymatuyama@yahoo.co.jp (H.T.); 2Department of Nutrition and Metabolism, National Institute of Health and Nutrition, National Institutes of Biomedical Innovation, Health, and Nutrition, 1-23-1 Toyama, Shinjuku-ku, Tokyo 162-8636, Japan; 3Graduate School of Sports and Health Science, Fukuoka University, 8-19-1 Nanakuma, Jonan-ku, Fukuoka 814-0180, Japan; lrietty.tl@gmail.com; 4Graduate School of Frontier of Biosciences, Osaka University, 1-17 Machikaneyama, Toyonaka, Osaka 560-0043, Japan; ryoma100034@gmail.com; 5Department of Food and Health Sciences, Prefectural University of Kumamoto Faculty of Environmental and Symbiotic Sciences, 3-1-100 Tsukide, Higashi-ku, Kumamoto 862-8502, Japan; eyoshi@pu-kumamoto.ac.jp

**Keywords:** energy balance, physical activity, appetite, energy expenditure, energy intake

## Abstract

We aimed to investigate the effects of a wide range of daily physical activity (PA) levels on energy balance (EB), energy intake (EI), and appetite. Nine young men completed three different PA levels in a metabolic chamber in a random order: (1) no exercise (Low-PA); (2) 25 min walking seven times (Mid-PA); and (3) 25 min running seven times (High-PA) within a 24 h period. Interval exercise (25 min exercise and 35 min rest) was performed three times in the morning and four times in the afternoon. The exercise intensities were 21.6% and 53.7% $\dot{V}$O_2_ peak for the Mid-PA and High-PA days, respectively. Participants were served three standardized meals and a buffet for dinner. The 24 h EB was calculated as 24 h energy expenditure (EE) minus 24 h EI. The 24 h EEs for the Low-PA, Mid-PA, and High-PA days were 1907 ± 200, 2232 ± 240, and 3224 ± 426 kcal, respectively, with significant differences observed among the three conditions (*p* < 0.01 for Low-PA vs. Mid-PA, Low-PA vs. High-PA, and Mid-PA vs. High-PA, respectively). The 24 h EIs for the Low-PA, Mid-PA, and High-PA days were 3232 ± 528, 2991 ± 617, and 3337 ± 684 kcal, and were unaffected by PA levels (*p* = 0.115). The 24 h EBs were 1324 ± 441 kcal (Low-PA), 759 ± 543 kcal (Mid-PA), and 113 ± 430 kcal (High-PA), with significant differences observed between Low-PA vs. Mid-PA (*p* = 0.0496), Low-PA vs. High-PA (*p* ≤ 0.01), and Mid-PA vs. High-PA (*p* = 0.017) conditions. The EB in the Low-PA group was the highest of the three conditions. Appetite perception did not differ among the study days, however there was an interaction trend (*p* = 0.078, time × condition). Thus, significantly different daily PA did not affect 24 h EI, however markedly affected 24 h EB, implying that EB is not automatically matched during a single day.

## 1. Introduction

Body weight increases in response to a daily energy surplus (when daily energy intake [EI] exceeds energy expenditure [EE]). Sedentary behavior is a common feature of modern lifestyles (illustrated, for example, by the time spent driving, watching television, and working using computers, especially in developed countries) and predisposes people towards a state of energy surplus due to low EE.

Ideally, EI and EE are similar to ensure homeostasis and a stable body weight [1]. However, individuals who undertake low levels of physical activity (PA) appear to suffer a loss of appetite control [2,3]. In 1956, Mayer et al. evaluated the relationship between different workloads, comparing sedentary and heavy physical work with respect to EI in a cross-sectional study [4]. The results showed a J-shaped relationship between PA and EI. Interestingly, EI was higher in inactive workers, despite their lower PA, which is indicative of a state of positive energy balance, and such an inactive lifestyle would also lead to an increase in body weight. In contrast, in more active workers, EI increased linearly as workload increased, however body weight was stable, meaning that EE and EI were closely matched. A recent study [5] and a systematic review [6] replicated this J-shaped relationship. Thus, an inactive lifestyle dysregulates dietary intake such that EE and EI are dissimilar, and an energy imbalance is created.

Although many previous studies have evaluated the effects of acute exercise on appetite and EI [7], surprisingly few have investigated the effects of exercise or activity level on energy balance (EB) experimentally. Stubbs et al. evaluated the influence of medium–low (1.4 × resting metabolic rate [RMR]) and medium–high (1.8 × RMR) PA levels on EI in subjects with free access to food, assessing EE for seven consecutive days using whole-body indirect human calorimetry (HC) [8]. Although EI was similar under the two PA conditions, in contrast to the findings of Mayer’s study [4], EB under medium–low activity conditions was higher than that under medium–high activity conditions. Stubbs et al. selected 24 h EEs of 1.4 and 1.8 × RMR because the general population typically expends energy equivalent to a range of 1.4–1.8 × RMR daily [9]. However, the EE of many individuals lies outside the 1.4–1.8 × RMR range and therefore, the effects of a wider range of PA requires investigation if the results of Mayer’s cross-sectional study are to be validated experimentally. As stated above, many previous studies have determined the effects of acute exercise on EI within a few days and concluded that acute exercise is effective at producing a short-term energy deficit, especially within a single day [7], however, in general, EB has not been quantified by measuring both EE and EI.

To our knowledge, no studies have investigated the influence of a wider range of PA levels over the course of a day on EB. Thus, the purpose of this study was to investigate the effects of three disparate PA levels on EB, calculated using HC, EI, and appetite measurements, under ad libitum feeding conditions. We hypothesized that a high level of PA would be associated with zero EB and a low PA level would result in a positive EB.

## 2. Materials and Methods

### 2.1. Study Participants

Nine healthy male university students who did not undertake physical exercise more than once a week participated in the present study. The participants were recruited mainly via the laboratory information board at the Faculty of Sports and Health Science of Fukuoka University. The required sample size was calculated to be six using an α of 0.05, a β of 0.80, and EB data obtained in a previous study [8], using G*power Version 3.1.9 software (Dusseldorf University, Düsseldorf, Germany). The characteristics of the participants are shown in Table 1. All participants gave their written informed consent after reviewing the purpose, methods, and significance of the study. The inclusion criteria were that participants should be non-smokers, have a body mass index (BMI) of <30 kg/m^2^, not have diabetes or cardiovascular disease, and not be taking medication or antioxidant supplements. The study was approved by the Ethics Committee of Fukuoka University (No. 150703). The study was carried out in accordance with the principles of the Declaration of Helsinki (1975). The protocol was registered with the University Hospital Medical Information Network Clinical Trial Registry (UMIN000032913).

### 2.2. Experimental Design

The participants underwent baseline testing and then spent 24 h in a calorimeter on three occasions. Baseline testing consisted of an aerobic capacity test and the measurement of body composition. At least five days after this, the participants underwent their first trial in the calorimeter. They exercised at three different intensities in the calorimeter in a random order. These levels were (1) sedentary (Low-PA: low level of physical activity), (2) seven × 25 min walking (Mid-PA: intermediate level of physical activity), and (3) seven × 25 min jogging (High-PA: high level of physical activity), and the trials were separated by seven days. The participants were asked not to perform vigorous exercise for three days beforehand and to eat the same meals on the day before each experimental day.

The schedule of the experimental days is shown in Figure 1. On the experimental days, the participants came to the laboratory at 08:20 and entered the calorimeter at 08:30. Breakfast and lunch were at 09:30 and 13:30, respectively, and each lasted 20 min. Dinner was served between 19:30 and 23:00 and consisted of a standard meal and a buffet-style meal. The participants went to bed at 00:00, were woken at 08:00, and left the metabolic chamber at 09:20.

On two of the three test days, PA was performed for seven × 25 min between breakfast and dinner in the calorimeter (total period of PA = 175 min). The walking exercise intensity was at 50% of the lactate threshold (LT) intensity and the jogging exercise intensity was at the LT intensity. Heart rate (HR) was measured and perceived exertion (RPE; assessed using the Borg scale (6–20) [10]) was rated during the last minute of PA. The number of steps were counted using a triaxial accelerometer (Actimarker EW4800; Panasonic Electric Works, Osaka, Japan) which was worn at waist level during stays in the calorimeter.

On the study days, on the Low PA day, and when they were not exercising in the calorimeter on the Mid- and High-PA days, the participants were asked to remain seated on a chair, rather than lying down on the bed. They were permitted to browse the internet, read books, watch TV, and undertake routine daily activities (for example, using the bathroom and brushing their teeth).

### 2.3. Baseline Testing

#### 2.3.1. Aerobic Capacity Exercise Test

The participants fasted for 12 h before baseline testing when they completed a multi-stage aerobic capacity test on a treadmill [11]. This test had two purposes: (1) to determine maximal oxygen consumption ($\dot{V}$O_2_ peak); and (2) to determine their LT at running speed. This value was then used to calibrate subsequent exercise levels. After a 1 min rest, the multi-stage exercise tolerance test was then started at a running speed determined by each subject’s fitness level, and this was increased by 10 m/min every 4 min, with 1 min rest sessions between each bout.

Blood lactate concentration (LA, Lactate Pro 2LT-1730 Arkray, Kyoto, Japan) was measured immediately after each exercise bout until a concentration of 4 mmol/L was reached. HR (Polar RS800CX) was measured during the final 30 s of each stage. Participants were asked for their RPE after each stage. Once their LA value exceeded 4 mmol/L, the treadmill incline was increased by 2% per min until the participant was completely exhausted. During the exercise test, the concentrations of the expired gases were measured every 12 s using a mixing chamber method and a mass spectrometer calibrated for respiratory analysis (ARCO-2000, Arco System, Chiba, Japan). The $\dot{V}$O_2_ peak value accepted was the highest volume of oxygen consumption during 1 min of the aerobic capacity test. The %$\dot{V}$O_2_ peak at the LT was calculated using the linear relationship between running speed and oxygen consumption ($\dot{V}$O_2_ at LT/$\dot{V}$O_2_ peak × 100%).

#### 2.3.2. Body Composition

Body composition was estimated using the underwater weighing method and body density was calculated after correction for residual air using the O_2_ re-breathing method. Body fat percentage was calculated using a standard formula [12]. Body mass (BM) was measured using a calibrated balance beam scale (Shinko Denshi Vibra Co., Ltd., Tokyo, Japan) to the nearest 0.01 kg, with the subjects wearing only light undergarments. Fat mass (FM) and fat-free mass were calculated using the formulae BM × body fat percentage/100% and BM−FM, respectively. Height was measured to the nearest 0.1 cm using a stadiometer.

### 2.4. Testing during the Interventions

#### 2.4.1. Heart Rate and Perceived Exertion Rating

HR was measured using an HR monitor (Polar RS800CX) during the final 30 s of the first exercise bout (at 10:00–10:25) during the Mid-PA and High-PA trials. RPE was also recorded after the first exercise bout.

#### 2.4.2. Details of the Meals Consumed

Nutritional information for the standardized meals (breakfast and lunch) consumed in a day is shown in Table 2. Dinner consisted of a standardized meal and an ad libitum buffet meal (Appendix A). The EI (kcal) associated with the standardized meals was 0.85 × basal metabolic rate (RMR), estimated using a published RMR equation for Japanese people [13]. The energy content of the standardized meals was divided up as follows: breakfast, 40%; lunch, 40%; and dinner, 20%. Participants were permitted to eat as much or as little of each available item in the buffet as they wished and were told that they did not have to eat everything that was served. After the dinner period, the remaining food was immediately collected from the calorimeter, the rice remaining in the bowl was weighed, and the amounts of residual food and food intake were calculated by the managing dietician. Water was the only drink permitted.

#### 2.4.3. Assessment of Appetite

Hundred-millimeter visual analogue scales (VAS) were used throughout each day of the study to evaluate appetite perception (hunger, fullness, and appetite), which were defined as previously [14]. “Appetite” was defined as a sensation related to the maintenance of eating, often a desire for something specific; “hunger” was defined as a nagging, irritating feeling that signified food deprivation to the extent that the participant wished to eat; and “fullness” was defined as the sensation of the degree of stomach filling.

Participants placed a mark on the line at the point that best matched their appetite at the time. When participants were asked “How strong is your desire to eat?”, if the answer was “No desire to eat at all”, participants made a mark at 0 mm on the line. If the response was “I feel a strong desire to eat”, they made a mark at 100 mm on the line. In the same way, their degree of hunger and fullness were rated. For hunger, in response to “How hungry do you feel?”, “Not hungry at all” was rated at 0 mm, and “I have never been more hungry” was rated at 100 mm. For fullness, in response to “How full do you feel?”, “Not at all full” was rated at 0 mm and “Totally full” was rated at 100 mm. VAS scores were obtained at 8:30, 9:30, 10:00, 10:30, 11:30, 13:30, 14:30, 15:00, 15:30, 17:30, 19:30, 20:30, 21:30, and 24:00 (14 times during each trial day).

#### 2.4.4. Whole-Body Indirect Calorimetry

HC was used to measure O_2_ consumption and CO_2_ production in a room-size indirect calorimeter (Fuji Medical Science Co., Ltd., Chiba, Japan), as previously described [15]. The concentrations of O_2_ and CO_2_ in the outgoing air were measured using online process mass spectrometry (VG Prima δB, Thermo Electron Co., Winsford, UK). The temperature and relative humidity in the calorimeter were maintained at 25 °C and 50%, respectively, and the air in the calorimeter was removed at 80 L/min. VO_2_ and VCO_2_ were calculated using the Brown algorithm; EE was calculated using VO_2_, VCO_2_, and Weir’s equation (EE = 3.9 × VO_2_ (L) + 1.1 × VCO_2_ (L)) [16]; and the respiratory exchange ratio (RER) was calculated as VCO_2_/VO_2_. The accuracy of the calorimeter for the measurement of EE, determined repeatedly using a 3 h alcohol combustion test, was 99.7% ± 1.5% for O_2_ consumption and 100.1% ± 1.6% for CO_2_ production during the study. PA level was calculated as: PA level = 24 h minus EE/RMR. RMR was estimated using a previously published equation [13].

Net exercise-induced EE during Mid-PA (walking) and High-PA (jogging) was calculated as: EE for 1 h, including the initial period of exercise minus EE during the equivalent Low-PA period.

### 2.5. Statistical Analysis

Data are reported as mean ± SD. Step count, BM, 24 h EE, 24 h EI, 24 h RER, buffet meal intake, 24 h EB (24 h EI until 24 h EE), PA levels, HR, and RER during exercise were compared among the three PA levels using one-way repeated ANOVA and 95% confidence intervals (95% CIs), and *post hoc* testing using the Bonferroni method was then applied (SPSS version 23, IBM Corporation, USA). Each effect sizes (ES) were calculated using G*power Version 3.1.9 software.

Student’s paired *t*-test was used to compare 24 h EI and 24 h EE under each condition and to compare the values of exercise speeds, exercise intensity, total net exercise EE during exercise, and RPE after a bout of exercise during the Mid-PA and High-PA trials.

Two-way ANOVA with repeated measures (measurement parameter × time) was used to analyze differences in the scores for hunger, fullness, and appetite among the three trial days. Further, *p* < 0.05 was considered to represent statistical significance.

## 3. Results

All nine participants completed the three 24 h trials in the calorimeter. BM was not significantly different between the study days (*p* = 0.447, ES = 0.326).

### 3.1. Response to Exercise

Participants walked or ran for seven bouts of 25 min on the treadmill between 10:00 and 17:30. Table 3 shows the exercise conditions and responses on the Mid-PA and High-PA days. Average values of seven times were calculated for 15 min during exercise (5–20 min each exercise duration) for exercise intensity (%VO_2_ peak) and RER are also shown in Table 3. There was a significant difference among trials in HR (main effect of trial, ES = 5.262, *p* < 0.001), with post-hoc tests indicating higher HRs for High-PA vs. Low-PA (ES = 6.783, *p* < 0.001), High-PA vs. Mid PA (ES = 4.253, *p* < 0.001), and Mid-PA vs. Low-PA (ES = 2.513, *p* < 0.001) conditions, however not RER (ES = 0.290, *p* = 0.523) between exercise days and the resting day during equivalent time periods.

### 3.2. Twenty-Four-Hour Energy Expenditure and Respiratory Exchange Ratio in the Metabolic Chamber

The values of 24 h EE and RER are shown in Table 4 and Figure 2. The 24 h EEs differed significantly among the trials (main effect of trial, ES = 4.592, *p* < 0.001) with post-hoc tests indicating higher 24 h EEs for High-PA vs. Low-PA (ES = 4.381, *p* < 0.001), High-PA vs. Mid PA (ES = 4.348, *p* < 0.001), and Mid-PA vs. Low-PA (ES = 3.588, *p* < 0.001) conditions. 

The 24 h RERs were significantly different among trials (main effect of trial, ES = 1.310, *p* < 0.001) with post-hoc tests indicating higher 24 h RERs for Low-PA vs. Mid-PA (ES = 1.463, *p* = 0.007) and Low-PA vs. High-PA (E S= 1.405, *p* = 0.009) conditions, however no significant difference for Mid-PA vs. High-PA (ES = 0.555, *p* = 0.403) conditions.

PA levels, which means 24 h-EE/RMR values, were 1.26 ± 0.07, 1.47 ± 0.10, and 2.13 ± 0.21 for Low-PA, Mid-PA, and High-PA trials, respectively. These values were significantly different among trials (main effect of trial, ES = 5.003, *p* < 0.001), with post-hoc tests showing higher 24 h PA levels in High-PA vs. Low-PA (ES = 4.762, *p* < 0.001), High-PA vs. Mid PA (ES = 4.827, *p* < 0.001), and Mid-PA vs. Low-PA (ES = 3.598, *p* < 0.001) conditions.

The 24 h EEs for the parts of each study day are shown in Table 4. The exercise periods (9:00–19:30) demonstrated higher EE during the Mid-PA and High-PA trials than during the Low-PA trial, however during the dinner and sleep periods, it was not significantly different among the study days (Table 4). For RER, significant differences in all parts were observed for the main effect of trial (Table 4), with post-hoc tests indicating lower RER in Mid-PA vs. High-PA from 9:00–13:29 (ES = 1.585, *p* = 0.004), High-PA vs. Low-PA and Mid-PA from 19:30–23:59 (High vs. Low: ES = 1.909, *p* = 0.0013; High vs. Mid: ES = 1.936, *p* = 0.0012), and High-PA vs. Low-PA from 0:00–7:59 (ES = 1.430, *p* = 0.008). 

### 3.3. Energy Balance and Energy and Macronutrient Intake

The mean EIs and macronutrient (protein, fat, and carbohydrate) intakes from the buffet meal are summarized for each study day in Table 5. There were no significant differences in 24 h EI, EI, or absolute macronutrient intake from the buffet meal among the three study days. However, the percentages of macronutrient intake were significantly different (Table 5). Post-hoc tests only showed that % protein for Low-PA was lower than Mid-PA (ES = 1.033, *p* = 0.044), however other post hoc tests showed no difference between the trials.

For 24 h EBs, the 24 h EE and EI were significantly different on Low-PA (95% of CI = 986–1663 kcal, ES = 3.007, *p <* 0.001) and on Mid-PA days (95% of CI = 341–1176 kcal, ES = 1.397, *p <* 0.001), however not on the High-PA days (95% of CI = −218–444 kcal, ES = 0.262, *p =* 0.454). Therefore, 24 h EB was significantly different among the three study days. The significant differences were found among trials (main effect of trial, ES = 1.862, *p* < 0.001) with post hoc tests indicating higher 24 h EBs for High-PA vs. Low-PA (ES = 3.339, *p* < 0.001), High-PA vs. Mid-PA (ES = 1.247, *p* = 0.017), and Mid-PA vs. Low-PA (ES = 1.007, *p* = 0.0496 (Figure 2). 

### 3.4. Appetite Perception

Figure 3 shows the ratings for hunger, fullness, and appetite during the experimental days. Two-factor ANOVA revealed a main effect of time (*p* < 0.001), however an effect of PA level was not shown for each hunger, fullness, and appetite rating. There was no significant interaction (PA level × time) among the PA levels with regard to hunger, fullness, and appetite, however there was a trend for a difference in appetite (*p* = 0.078, PA level × time). 

## 4. Discussion

The purpose of this study was to investigate the effects of different levels of PA, from sedentary to a high level of PA, during a single day, on EB, EI, and the appetite of young, healthy men. The primary findings were that a large range of PA levels (RMR × 1.26–2.10) in a single day affected 24 h EB, that a substantial positive EB (mean + 1272 kcal) developed on the sedentary day, and that as the PA level increased, the difference in EB moved closer to zero because the PA level did not affect absolute ab libitum EI. Our results indicate that even high PA levels are not always negative for EB, and a wider range of PA levels did not influence the appetite perceptions and absolute EI.

### 4.1. Physical Activity Level and Energy Balance

To our knowledge, only one previous study has evaluated Mayer’s findings experimentally, examining the effects of different physical activity levels on EB by quantifying EI and EE. Stubbs et al. reported that moderate physical activity (RMR × 1.4) induced a higher positive EB by HC than more intense physical activity (RMR × 1.8) when food was available ad libitum [8]. The present study was shorter in duration than that conducted by Stubbs (one day vs. one week), however it evaluated the effects of more disparate PA levels on EB than Stubbs’s study. We have shown that minimal activity is associated with a more positive EB than when more PA is undertaken, which is consistent with the results of the previous study [8], and previous studies have also shown that low PA would likely be associated with positive EB for at least a few days [17,18].

When the participants undertook High PA (EE equivalent to RMR × 2.1), we found that EB was close to zero, with little difference between the mean EI and EE. In general, EI and EE should be similar to maintain homeostasis and a stable body weight [1]. During the day of High PA, however, we believe that this coupling between 24 h-EI and EE happened by chance, rather than because of a homeostatic mechanism. Stubbs et al. showed that PA resulting in EE equivalent to RMR × 1.8 was associated with a low positive EB using very similar methodology (HC and a buffet meal test) to the present study (EI > EE) [8]. Ad libitum feeding results in a “buffet effect”, whereby dietary variety promotes overconsumption [19]. The present findings indicate that the EE associated with more than twice the normal PA level per day may be required to maintain EB under ad libitum feeding conditions.

### 4.2. The Effect of Physical Activity and Exercise on Energy and Macronutrient Intake

Taking our data and those of previous studies together, short-term EB (≤1 week) is probably determined by EE (mainly resulting from exercise or other physical activity) under normal dietary conditions. The reason for this is that exercise or PA has little effect on EI during a single day if bouts are limited to ≤60 min of moderate intensity exercise [7,20,21], even if the exercise undertaken is substantial (≤100 min [22,23] or 120 min [8,24,25]), which is the case if this amount of exercise is undertaken for two consecutive days [26] or for a week [8,17,18,21,24]. Furthermore, a previous review stated that EI does not increase within 16 days of daily exercise [27]. Although many previous studies involved continuous exercise for up to 60 min, the present exercise protocol involved frequent shorter exercise bouts, however the total duration of the exercise was 175 min, which is longer than in the previous studies. Studies involving cumulative exercise, using protocols such as two bouts of 50 min (morning and afternoon) [23] or three bouts of 40 min during one day [8,24], were aerobic and did not affect EI. If the present study had used a single prolonged period of exercise, the results would probably have been quite different from those obtained using cumulative exercise because of a difference in the time taken to reach fatigue, which is known to influence the secretion of appetite-regulating hormones and, thus, the degree of exercise-induced anorexia [28]. The present study divided the total period of exercise into seven bouts, with each being relatively short, in an attempt to avoid the direct influence of prolonged exercise on appetite and to model the daily activity pattern of people undertaking heavy labor as closely as possible. Both the present and previous findings indicate that cumulative high-volume physical activity during a single day does not influence EI that day.

The lack of influence of large differences in PA on EI in the present study may have been due to the use of a buffet meal test in the present study. The variety of items available in the buffet meal and the long duration of availability of the food may have induced overeating on each of the days of the study. In advance, we predicted that the use of such a duration and style of buffet meal would be more likely to elicit differences in EI, alongside the disparate levels of PAs, however in fact the opposite appears to have occurred. Although the effect of overfeeding is unclear, one previous study has shown that there is no relationship between meal duration and intake in a fast-food restaurant [29]. In our view, even if the use of a buffet meal represents a confounding factor, the present findings are consistent with a large difference in PA (sedentary vs. ~3 h of PA) in a single day not influencing EI.

Some previous reports have stated that as PA increases, so does EI [2,4,6], however this correlation probably requires long-term differences in PA levels to appear [24,30,31]. During the short duration of the present and previous experimental studies, PA level does not appear to impact food consumption; instead, participants tend to continue their habitual food intake patterns. Thus, it appears that compensatory changes in EI resulting from greater EE may take longer than one day to develop, however may be of short duration (≤1 week), such that EB is likely to be determined principally by EE.

As stated above, many previous studies have evaluated the effects of a bout of exercise on EI and have reported no, or a trivial, effect on subsequent EI [7]. It was concluded that performing a single bout of exercise could induce negative EB on the day of exercise because no effect of exercise on EI was identified when exercising and sitting were compared. However, few studies have calculated EB following measurement of both EI and EE. In fact, negative EB is not always generated on the day of exercise. For example, the choice of a highly energy-dense food after exercise is often associated with a positive EB [32,33,34], and ad libitum feeding is also likely to lead to overconsumption [35]. Thus, acute energy deficit from increased PA would only be achieved when the daily meal pattern is considered without high energy dense food or the buffet meal.

Thus, one of the strong points of our study is that we quantified EB in association with both exercise and ad libitum feeding during a single day and have shown that even short periods of significant PA do not always induce negative EB in single day. This may be one reason that people who exercise, however do not modify their diet, fail to lose weight.

A previous systematic review concluded that there is no influence of short-term exercise on macronutrient intake [20]. With regard to longer periods of exercise, King et al. reported that a prolonged period of exercise (90 min of running) had no effect on macronutrient intake during a buffet-style meal [22], whereas Kojima et al. reported that carbohydrate intake during a buffet meal following a 20-km run was higher than after no exercise [36]. In contrast, in the present study of the effect of cumulative exercise, the percentages of macronutrients consumed at the buffet meal were slightly different on the three study days, with carbohydrate intake on the Low-PA day being higher than on the Mid and High-PA days. Although the reason for these differing findings is unclear, differences in the buffet meal test or the background of the subjects may have played a role.

### 4.3. The Effect of Physical Activity and Exercise on Appetite Perception

A number of previous studies have reported that people feel less hungry during and soon after a bout of exercise, especially if the exercise intensity is equivalent to >60% of maximal oxygen uptake [22,33,37]. In the present study, the exercise intensity on the High-PA day was 53.7% ± 10.4% of $\dot{V}$O_2_ peak and there were no differences in appetite perceptions or trends for mean appetite. At a glance, lower average values resulted from 15:30–17:30 on High-PA trials than Mid- and Low-PA trials because the fatigue induced by repeated running may be cumulative. A number of previous studies reported that VAS score is not necessarily predictive of EI [38,39,40,41]. Indeed, a review article indicated that acute exercise does not increase hunger or EI in normal-weight individuals, and our results support this view and extend its interpretation, as even when High-PA exercise was performed for the entirety of the day, it did not influence appetite perceptions and EI [27].

If we extrapolate our findings to comment on the significance for normal lifestyles, our study may mirror the differences in EB associated with shifts in the lifestyle of people, such as sedentary office workers, at the weekend, which could lead to body weight gain [42]. An energy imbalance may be the result of higher EI and lower PA on the weekend than during the week [42,43]. The present data imply that if someone does not undertake exercise on a single day, their EI is not adjusted to compensate for the lower EE. Instead, there is the potential for a positive EB of >1000 kcal that day. Furthermore, if this “weekend lifestyle” is a regular event, it is likely to lead to a cumulative energy surplus and body weight gain in the longer term.

There are some notable limitations to the present study. First, the participants were comparatively healthy young men and were few in number. In the future, we plan to evaluate the responses in women, people with obesity, middle-aged men, and older men, as well as in larger numbers. Second, the study was only 24 h long, which appears insufficient to generate a difference in EI. Further research is needed to evaluate the effects of differences in PA on EB and EI over a longer period.

## 5. Conclusions

The present study has shown that PA levels equivalent to RMR × 1.26 and 2.13 during a single day do not influence absolute 24 h EI, however do affect 24 h EB. This implies that the difference in PA within a single day is the main determinant factor of EB and that EI is not automatically matched to PA level. Depending on the diets, even high PA or acute exercise in a day does not always induce a negative EB, and this may be one of the reasons for it not being an effective means of losing weight. The present study also suggests that a lack of exercise during a single day has the potential to lead to a high positive EB and a risk of body weight gain in the future.

## Figures and Tables

**Figure 1 nutrients-11-00690-f001:**
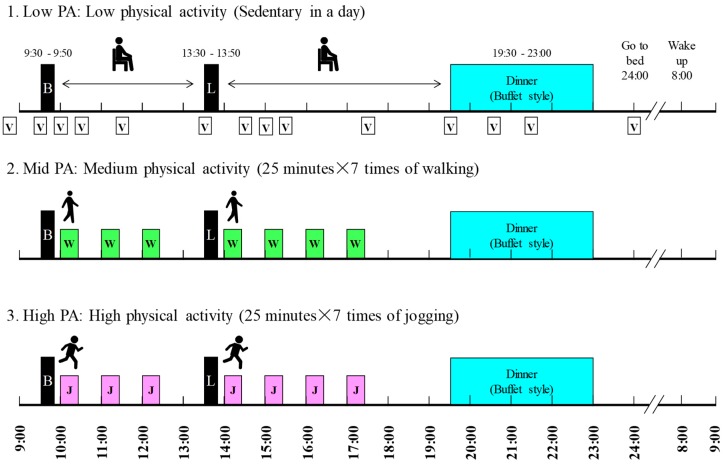
Experimental protocol. Black boxes indicate the meal times. Breakfast and lunch were standardized meals and were provided for 20 min each. Dinner was a buffet meal, provided for 3.5 h. White (walking) and gray (running) bars indicate the periods of exercise, each of which lasted 25 min. The total exercise duration for the Mid-PA and High-PA levels was 175 min. Energy expenditure was measured for 24 h (09:00–09:00) on each study day. Abbreviations: B = breakfast, L = lunch, W = walking, J = jogging, V = visual analogue scale rating of appetite.

**Figure 2 nutrients-11-00690-f002:**
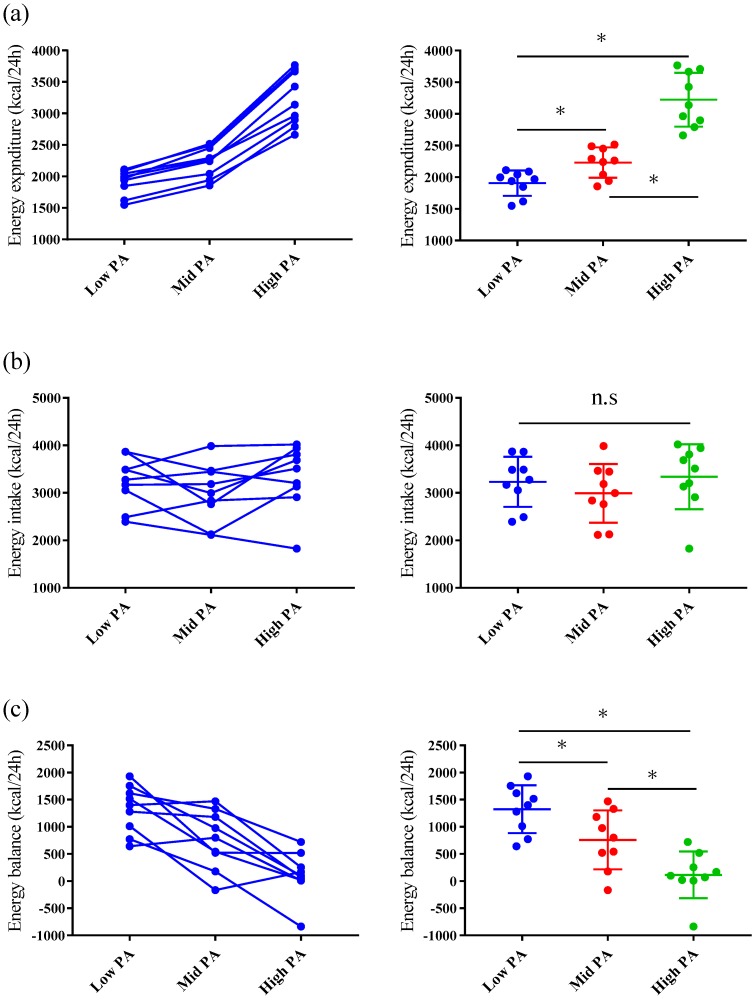
Comparison of energy expenditure (**a**), energy intake (**b**), and energy balance (**c**) on each study day. The panels on the left show the differences among the three study days for each individual. The panels on the right show the variation in the data on each day (mean and SD). * *p* < 0.05.

**Figure 3 nutrients-11-00690-f003:**
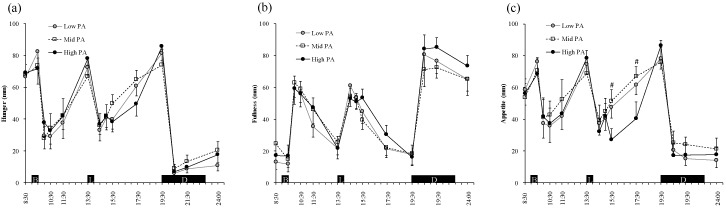
Perceptions of hunger (**a**), fullness (**b**), and appetite (**c**) during the three study days. Symbols: Low-PA (●), Mid-PA (□), and High-PA (●). Values are the means for the participants (*n* = 9).

**Table 1 nutrients-11-00690-t001:** Participant baseline characteristics (*n* = 9).

Age (years)	22.2 ± 1.6
Height (cm)	170.8 ± 3.6
Body mass (kg)	64.7 ± 9.8
BMI (kg/m^2^)	22.1 ± 3.0
Body fat (%)	18.2 ± 6.3
Fat mass (kg)	12.0 ± 5.2
Fat-free mass (kg)	52.6 ± 6.9
$\dot{V}$O_2_ peak (mL/kg/min)	50.0 ± 4.3

BMI = body mass index, $\dot{V}$O_2_ peak = peak oxygen uptake.

**Table 2 nutrients-11-00690-t002:** Energy intake, nutrient intake, and components of the three standardized meals on each experimental day (*n* = 9).

	Breakfast	Lunch	Dinner
Energy Intake (kcal/day)	545 ± 52	531 ± 45	266 ± 19
Macronutrient (g)			
Protein	12.5 ± 2.5	17 ± 1.4	6.1 ± 0.5
Fat	17.4 ± 1.5	15 ± 1.2	8.6 ± 0.7
CHO	86.8 ± 8.9	81 ± 7.4	41.1 ± 3.7
Macronutrient (%)			
Protein	9.0 ± 1.1	12.8 ± 0.4	9.2 ± 0.2
Fat	28.7 ± 3.3	25.6 ± 0.2	29.3 ± 1.7
CHO	62.3 ± 2.3	61.5 ± 0.3	61.4 ± 1.8
Meal composition	Granola with milk	Cod roe pasta (FZM) and a bread roll	Curry and rice

Values are mean ± SD. CHO, carbohydrate; FZM, frozen meal.

**Table 3 nutrients-11-00690-t003:** Total steps taken on each study day and exercise responses during each study day (*n* = 9).

Variable	Low_PA	Middle_PA	High_PA	*p*	Low vs. Mid PA	Low vs. High PA	Mid vs. High PA
	mean ± SD	mean ± SD	mean ± SD		95% CI	95% CI	95% CI
Steps (day)	275 ± 306	19,219 ± 2043	28,644 ± 1043	<0.001	16,855, 21,033	27,323, 29,414	7104, 11,745
Exercise Speeds (m/min)	-	60 ± 13	120 ± 26	<0.001	-		50, 70
Exercise intensity (%VO_2_ peak)	-	21.6 ± 3.7	53.7 ± 10.4	<0.001	-		11.8, 25.9
HR (beats/min)	69 ± 6	93 ± 12	145 ± 12	<0.001	15, 34	65, 87	39, 64
RER during exercise	0.859 ± 0.049	0.873 ± 0.046	0.862 ± 0.035	0.523	−0.036, 0.065	−0.027, 0.034	−0.046, 0.024
Total net Ex EE (kcal)	-	344 ± 66	1261 ± 254	<0.001	-		745, −1089
RPE	-	7.6 ± 2.1	11.7 ± 1.7	0.001	-		2.3, 5.9

HR, heart rate; PA, physical activity; RER, respiratory exchange ratio; Ex EE, energy expenditure due to exercise; RPE, rating of perceived exertion. Total net Ex EE was calculated as Mid or High-PA EE minus Low-PA EE between 10:00 and 17:30, which included the periods of exercise.

**Table 4 nutrients-11-00690-t004:** Energy expenditure, energy balance, and respiratory exchange ratio during the three study days (*n* = 9).

Time of Day	Low_PA		Middle_PA		High_PA		*p*	Low vs. Mid PA	Low vs. High PA	Mid vs. High PA
**Enery expenditure (kcal)**	mean ± SD	95% CI	mean ± SD	95% CI	mean ± SD	95% CI		95% CI	95% CI	95% CI
9:00–13:29 (Morining–before Lunch)	388 ± 35 *^,#^	361, 414	539 ± 54 ^&^	497, 580	912 ± 127	814, 1010	<0.001	120, 182	414, 635	268, 480
13:30–19:29 (Lunch–before Dinner)	489 ± 56 *^,#^	445, 532	682 ± 82 ^&^	619, 745	1244 ± 184	1102, 1386	<0.001	147, 240	592, 919	431, 692
19:30–23:59 (Dinner–before bed)	427 ± 46	392, 463	414 ± 55	372, 457	440 ± 64	391, 489	0.111	−48, 23	13, 39	−14, 66
0:00–7:59 (Sleep)	526 ± 64	477, 575	520 ± 59	475, 566	547 ± 80	485, 608	0.158	−37, 26	−17, 58	−25, 77
**Respiratory exchange ratio**	mean ± SD	95% CI	mean ± SD	95% CI	mean ± SD	95% CI		95% CI	95% CI	95% CI
24-h RER	0.883 ± 0.034 *^,#^	0.858, 0.909	0.865 ± 0.034	0.839, 0.892	0.857 ± 0.025	0.838, 0.877	<0.001	−0.031, −0.006	−0.045, −0.007	−0.022, 0.006
9:00–13:29 (Morining–before Lunch)	0.844 ± 0.034	0.818, 0.871	0.835 ± 0.030 ^&^	0.812, 0.859	0.855 ± 0.024	0.836, 0.874	0.004	−0.025, 0.007	−0.006, 0.027	0.007, 0.033
13:30–19:29 (Lunch–before Dinner)	0.872 ± 0.030	0.848, 0.895	0.861 ± 0.031	0.837, 0.885	0.856 ± 0.021	0.839, 0.872	0.022	−0.027, 0.005	−0.033, 0.001	−0.02, −0.009
19:30–23:59 (Dinner–before bed)	0.891 ± 0.036 ^#^	0.863, 0.919	0.884 ± 0.036 ^&^	0.856, 0.912	0.859 ± 0.035	0.831, 0.886	<0.001	−0.019, 0.005	−0.050, −0.015	−0.038, −0.012
0:00–7:59 (Sleep)	0.912 ± 0.046 ^#^	0.876, 0.947	0.884 ± 0.055	0.841, 0.926	0.863 ± 0.037	0.835, 0.892	0.002	−0.060, 0.004	−0.082, −0.014	−0.056, 0.016

The Bonferroni post hoc test was applied. * *p* < 0.05 for Low vs. Mid; ^#^
*p* < 0.05 for Low vs. High; and ^&^
*p* < 0.05 for Mid vs. High-PA.

**Table 5 nutrients-11-00690-t005:** Energy and macronutrient intakes during the buffet meal on each study day (*n* = 9).

	Low_PA		Middle_PA		High_PA		*p*	Low vs. Mid PA	Low vs. High PA	Mid vs. High PA
	Mean ± SD	95% CI	Mean ± SD	95% CI	Mean ± SD	95% CI		95% CI	95% CI	95% CI
Energy intake during buffet meal (kcal)	1880±502	1494, 2266	1639 ± 576	1196, 2081	1985 ± 637	1496, 2475	0.1149	−797, 315	−224, 435	−178, 871
Protein (g)	72 ± 30	49, 95	68 ± 23	50, 86	83 ± 34	57, 108	0.1155	−29, 20	−6, 27	−5, 35
Fat (g)	66 ± 30	43, 88	61 ± 30	38, 85	74 ± 29	52, 97	0.150	−28, 19	−3, 20	−8, 34
Carbohydrate (g)	250 ± 52	211, 290	204 ± 72	149, 259	247 ± 82	183, 310	0.073	−112, 20	−52, 45	−27, 112
Protein (%)	15 ± 3	12, 17	17 ± 3	15, 19	17 ± 3	14, 19	0.050	0, 4	−1, 4	−3, 2
Fat (%)	30 ± 9	23, 37	33 ± 8	27, 40	34 ± 6	29, 38	0.045	−1, 8	−1, 8	−3, 4
Carbohydrate (%)	55 ± 11	46, 64	50 ± 9	43, 57	50 ± 8	44, 56	0.015	−10, 0	−11, 1	−5, 4

Values are presented as mean ± SD.

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
