# Peer review of "Effects of Different Physical Activity Levels during a Single Day on Energy Intake, Appetite, and Energy Balance: A Preliminary Study"

_nutrients, 2019, doi:10.3390/nu11030690_

Round 1

Reviewer 1 Report

Major concerns:

 (1) The authors could motivate the study better. Specifically, the authors should answer the question why the focus on 24-hour assessment (as opposed to multiple-day assessments) is needed and what their contribution is.

(2) The authors should describe in more detail what the instructions were when the participants did not exercise. Could they move around freely, or were they instructed to remain sedentary?

Minor concerns:

 Line 74           n = 9?

Line 180         9 or 8 participants?

Line 279-81    Sentences are unclear and need to be reformulated

Author Response

Response to Reviewer 1 Comments

Comments and Suggestions for Authors

Major concerns:

 (1) The authors could motivate the study better. Specifically, the authors should answer the question why the focus on 24-hour assessment (as opposed to multiple-day assessments) is needed and what their contribution is.

●Response:

Thank you for comment.

There were two principal reasons why we used a 24-hour assessment.

First, we thought that we should assess a wider range of physical activity to verify the J-shaped relationship experimentally, because the physical activity levels (PALs) studied by Stubbs et al. were 1.4 and 1.8 × resting metabolic rate (RMR), which were thought to be typical levels of PA in the general population, but these PALs did not influence energy intake (EI). We also believe that investigation of the effect of differences in PAL over multiple days is important, but we have initially focused on the effect of a larger difference in PAL on energy balance (EB) and energy intake (EI) in this preliminary study, and the next step will be to conduct a multiple-day study.

Second, many previous studies have evaluated the acute effects of exercise (physical activity) on appetite and EI, and a previous meta-analysis [1] and systematic review [2] concluded that acute exercise is effective at producing a short-term energy deficit in a single day. However, EB could not be quantified in these studies, because either energy expenditure (EE) or EI were measured, but not both. Therefore, it was unclear whether acute exercise always induces short-term negative EB in a single day, and we chose to focus on this question.

The present study did not demonstrate negative EB when exercise was performed for 175 min, because the use of a buffet meal may have caused overeating. Thus, acute exercise does not always generate negative EB in a single day, even if a PAL of ~2.1 × RMR is used. The difference of PA during a single day affected EB but did not affect absolute EI, implying that PA level is the main determinant of EB in the short term, because it does not affect EI. We think that this is a key finding of the study.

Nevertheless, in line with your comment, we propose to evaluate the impact of a multiple-day PA protocol in a follow-up study.

We have added text to discuss our objectives further in the Introduction (lines 63–67 and further discussion of these points to the Discussion (lines 300–313) and Conclusion (lines 380–386).

References

1.               Schubert, M.M.; Desbrow, B.; Sabapathy, S.; Leveritt, M. Acute exercise and subsequent energy intake. A meta-analysis. Appetite 2013, 63, 92-104, doi:10.1016/j.appet.2012.12.010.

2.               Donnelly, J.E.; Herrmann, S.D.; Lambourne, K.; Szabo, A.N.; Honas, J.J.; Washburn, R.A. Does increased exercise or physical activity alter ad-libitum daily energy intake or macronutrient composition in healthy adults? A systematic review. PloS one 2014, 9, e83498.

(2) The authors should describe in more detail what the instructions were when the participants did not exercise. Could they move around freely, or were they instructed to remain sedentary?

Response:

We thank the reviewer for this suggestion. We asked the participants to remain seated but not to lie down on the bed during the whole-body indirect human calorimetry. They were permitted to perform daily living activities (going to the bathroom and brushing their teeth), and could read and watch TV.

We have now added detail of the instructions given in lines 118–121.

Minor concerns:

Line 74           n = 9?

Line 180         9 or 8 participants?

Response: We are sorry for this confusion. There were nine participants, and we have corrected this information in the manuscript (lines 87 and 209).

Line 279-81    Sentences are unclear and need to be reformulated

Response: Thank you for pointing this out. We have edited the sentences and they have been checked by a native English speaker (lines 390-391).

Reviewer 2 Report

General comments

This paper addresses an important issue by examining the interaction between exercise, appetite and energy balance under extremely well-controlled and standardised conditions. The authors reported that increasing physical activity by accumulating bouts of structured exercise throughout the day did not lead to any compensatory changes in energy intake, resulting in a lower 24-h energy balance. Few studies have adopted the necessary measurements to address this question robustly and the current findings act to complement and extend the existing literature investigating the impact of acute exercise on appetite control outcomes. The experimental design has been well-conceived and are consistent with a rigorous data collection. However, I do have some minor and more extensive suggestions that the authors may wish to consider in revising the manuscript as outlined below:

Specific comments

Abstract

1.        It would be useful to include some additional details of the exercise sessions in the abstract (lines 19-20) in terms of the intensity of the exercise bouts and time interval between the 25 min bouts. It would also be useful to provide the mean(SD) values for 24 h energy intake in the results section of the abstract for completeness (lines 25-26).

Introduction

2.        Page 2, line 46: Previous research including those cited by the authors suggests that the relationship between energy intake and energy expenditure is ‘J-shaped’ not ‘U-shaped’.

Methods

3.        Page 2, lines 66-73: Was the sample size based on a power calculation and expected drop out rates? The  number of participants recruited to the study should be clarified. The text suggests 8 participants were recruited but the figures and tables suggest n = 9. It should be clarified if one participant dropped out and the reason for discontinuing participation. It would also be useful to provide a description of the habitual physical activity levels of the participants in this paragraph.

4.        Page 2, line 74: Information on fat mass (kg) and lean body mass (kg) should be included in the participant characteristics table as described on page 4, lines 127-128.

5.        Page 2, lines 79-82: What was the rationale for accumulating exercise bouts throughout the day rather than performing one continuous bout of exercise? This should be discussed in the paper to provide a rationale for accumulating exercise bouts throughout the day.

6.        Page 3, lines 87-88: Why did participants have access to the buffet meal for 3.5 h? Were the participants familiarised with the buffet meal before the trials? The provision of a wide array of food items available for a long duration (3.5 h) may promote overconsumption and reduce the ability to identify differences between the trials. This should be included as a discussion point in the manuscript.

7.        Page 4, lines 131-134: Why were heart rate and RPE only monitored during the final 30 s of the first exercise bout and not every exercise bout? This would have provided a more accurate insight into the physiological responses during the exercise bouts. Was the treadmill speed adjusted during exercise to ensure the participants were exercising at the target exercise intensity? It would be useful if additional details of the exercise sessions were included in Table 4 (e.g., exercise intensity (% peak VO2), respiratory exchange ratio, carbohydrate oxidation, fat oxidation). The net EE of exercise would be better presented as total net EE rather than an average of the 7 exercise bouts.

8.        Page 5, lines 147-153: It is unclear what the ‘appetite’ VAS scale is assessing and how this differs from hunger and fullness? Hunger and fullness are ratings of perceived appetite. The time points that appetite ratings were assessed should be described in the methods and outlined on Figure 1. Why was appetite not assessed on the morning of day 2?

9.        Page 5, lines 154-165: Did participants empty the bladder before entering the indirect calorimeter? Were urine samples collected to analyse nitrogen content during the day? This would enable the calculation of protein oxidation and subsequently protein balance which will provide a more accurate insight into energy and substrate balance over the course of the day.

10.     Page 5, lines 166-167: Why was net EE only calculated for 1 h rather than the entire exercise duration (i.e., 175 min)?

11.     Page 5, lines 168-178: It would be beneficial to include 95% confidence intervals and effect sizes to supplement the P values presented in the paper. 95% confidence intervals and effect sizes provide a better insight into main effects than more traditional dichotomous hypothesis testing (Hopkins et al. Med Sci Sports Exerc 2009;41(1):3-12; Shakespeare et al. Lancet 2001;357:1349-53). Alpha values only provide an arbitrary threshold, beyond which the findings are statistically significant, which is typically dependent on sample size. However, it is possible that a non-significant result may hide an important difference that could exist but was not detected. Confidence intervals provide a range of values within which the true effect is likely to lie and provide information on the size and direction of the treatment effect compared with the control. By supplementing confidence intervals with effect sizes, we are able to identify the magnitude of the effect.

Results

12.     Pages 5-7: The results section is slightly confusing in places and some re-wording is required to clarify the findings and confirm the trials that are being compared.

13.     Pages 6, lines 192-196: The authors could consider analysing the 24h EE in sub-sections throughout the day to examine changes in EE in different periods e.g., morning, afternoon, evening, sleep.

14.     Pages 6-7, lines 201-213: In addition to comparing energy balance between trials, details of 24-h substrate oxidation (i.e., protein, fat and carbohydrate) should be compared across trials. This is particularly relevant given the comparison of different exercise intensities which is likely to result in different patterns of substrate utilisation during exercise.

15.     Page 6: Figure 2 should be changed to present individual participant data with a line representing the mean rather than a plunger plot which can often be misleading (Drummond & Vowler 2011 Br J Pharmacol 163: 208-210; Weissgerber et al. 2015 PLOS Biol  13: e1002128; Weissgerber et al. 2016 PLOS Biol 14: e1002484). Similar plots could also be included for substrate balance i.e., protein, carbohydrate and fat balance.  

Discussion

16.     Pages 7-9: The discussion section would benefit from highlighting the wider implications of the findings for energy balance particularly in the opening and concluding paragraphs.

17.     Page 8, lines 269-270: Include a reference to support this point.

18.     Page 9, line 317: The mid PA trial resulted in a PA level equivalent to 1.47 x RMR not 1.3 x RMR. This should be updated.

19.     The paper is well written and concise, but there are some errors in grammar and word choice in places which should be addressed in a revised version of the paper.

Author Response

Response to Reviewer 2 Comments

Comments and Suggestions for Authors

General comments

This paper addresses an important issue by examining the interaction between exercise, appetite and energy balance under extremely well-controlled and standardised conditions. The authors reported that increasing physical activity by accumulating bouts of structured exercise throughout the day did not lead to any compensatory changes in energy intake, resulting in a lower 24-h energy balance. Few studies have adopted the necessary measurements to address this question robustly and the current findings act to complement and extend the existing literature investigating the impact of acute exercise on appetite control outcomes. The experimental design has been well-conceived and are consistent with a rigorous data collection. However, I do have some minor and more extensive suggestions that the authors may wish to consider in revising the manuscript as outlined below:

●Response:

We really appreciate your review, because we think that our revised manuscript has been substantially improved by addressing your suggestions and comments. The point-by-point responses and the changes made to the manuscript are described below.

Specific comments

Abstract

1.        It would be useful to include some additional details of the exercise sessions in the abstract (lines 19-20) in terms of the intensity of the exercise bouts and time interval between the 25 min bouts. It would also be useful to provide the mean(SD) values for 24 h energy intake in the results section of the abstract for completeness (lines 25-26).

●Response:

We thank the reviewer for these suggestions. We have added the requested information regarding the exercise protocol and the 24-h energy intakes on lines 21–23 and 26–27.

Introduction

2.        Page 2, line 46: Previous research including those cited by the authors suggests that the relationship between energy intake and energy expenditure is ‘J-shaped’ not ‘U-shaped’.

●Response:

Thank you for pointing this out. We have changed the text from “U-Shaped” to “J-Shaped” on lines 45 and 59.

Methods

3.        Page 2, lines 66-73: Was the sample size based on a power calculation and expected drop out rates? The number of participants recruited to the study should be clarified. The text suggests 8 participants were recruited but the figures and tables suggest n = 9. It should be clarified if one participant dropped out and the reason for discontinuing participation. It would also be useful to provide a description of the habitual physical activity levels of the participants in this paragraph.

●Response:

Thank you for your comments. We calculated the sample size for the present study using G*power software and the effect size reported by Stubbs et al. [2004], who studied individuals exercising in a similar way (lines 76–78). The calculated sample size was six, but we studied three different exercise intensities, whereas Stubbs et al. studied only two, and there was some risk of participants dropping out. Therefore, we chose to recruit nine subjects. Fortunately, however, there were no drop-outs. We have added this information on lines 74–79.

We had previously stated the number of the subjects as eight in the text, but the correct number was nine. We apologize for this error and are grateful for you pointing it out. It has now been corrected.

Finally, we were not able to assess the habitual PALs of the participants before and after the study. Instead, we have added a sentence regarding their background on line 74. However, we agree that this would be useful information, and in lieu we show their fitness level and body composition in Table 1. We plan to establish more clearly the habitual PALs of the participants in future studies.

4.        Page 2, line 74: Information on fat mass (kg) and lean body mass (kg) should be included in the participant characteristics table as described on page 4, lines 127-128.

●Response:

Thank you for your recommendation. This information has now been added to Table 1.

5.        Page 2, lines 79-82: What was the rationale for accumulating exercise bouts throughout the day rather than performing one continuous bout of exercise? This should be discussed in the paper to provide a rationale for accumulating exercise bouts throughout the day.

●Response:

The reason for the use of cumulative exercise bouts, rather than continuous exercise, was that if 175 min of continuous exercise was performed, EI or appetite would probably have been more strongly influenced by fatigue than would have been the case using cumulative exercise. One bout of very prolonged exercise is more likely to cause anorexia than cumulative exercise because of a difference in the level of fatigue induced, which is known to influence the secretion of appetite-regulating hormones [1]. We wished to focus on the effect of higher EE over a single day on EB and EI, rather than the effect of a bout of prolonged exercise, with the potential for an effect of fatigue on appetite. In summary, as far as we are aware, no previous study has evaluated the differential effects of prolonged continuous exercise and cumulative exercise on EI, but we think that these types of exercise may influence appetite differently.

We also wanted to model the type of daily activity pattern that might typically be present in manual laborers over the course of a day, which would tend to be intermittent, rather than continuous.

              We have added some discussion of this point to the Discussion on lines 300–313.

Reference

1.         Hazell, T.J.; Islam, H.; Townsend, L.K.; Schmale, M.S.; Copeland, J.L. Effects of exercise intensity on plasma concentrations of appetite-regulating hormones: Potential mechanisms. Appetite 2016, 98, 80-88, doi:10.1016/j.appet.2015.12.016.

6.        Page 3, lines 87-88: Why did participants have access to the buffet meal for 3.5 h? Were the participants familiarised with the buffet meal before the trials? The provision of a wide array of food items available for a long duration (3.5 h) may promote overconsumption and reduce the ability to identify differences between the trials. This should be included as a discussion point in the manuscript.

●Response:

Thank you for your comment.

The buffet meal was available for 3.5 h because we predicted that this long duration would help provoke a difference in EI and macronutrient intake due to the large differences in PA. However, no difference in EI was identified between the study days. Instead, as you have proposed, the novelty and long duration of availability of the buffet may have confounded our attempt to detect a difference in EI between the study days, due to overconsumption, a phenomenon that is known to be associated with the use of buffets, and which we had mentioned in our original manuscript. Although one previous study showed that meal duration does not affect the level of food intake [2], it is unclear whether in this instance the prolonged food availability lead to overeating and whether this may have reduced our ability to identify differences in EI between the study days.

Before the study began, we explained the use of a buffet meal test to the participants and provided a menu, and because it was used as part of a randomized cross-over study design, we do not think that an order effect is likely to have confounded the outcomes.

We have added some further discussion about the possibility that a long meal duration and the provision of a buffet might have induced overeating, thereby reducing any difference in EI between the study days, on lines 317–322.

Reference

2.         Brindal, E.; Wilson, C.; Mohr, P.; Wittert, G. Does meal duration predict amount consumed in lone diners? An evaluation of the time-extension hypothesis. Appetite 2011, 57, 77-79, doi:10.1016/j.appet.2011.03.013.

7.        Page 4, lines 131-134: Why were heart rate and RPE only monitored during the final 30 s of the first exercise bout and not every exercise bout? This would have provided a more accurate insight into the physiological responses during the exercise bouts. Was the treadmill speed adjusted during exercise to ensure the participants were exercising at the target exercise intensity? It would be useful if additional details of the exercise sessions were included in Table 4 (e.g., exercise intensity (% peak VO2), respiratory exchange ratio, carbohydrate oxidation, fat oxidation). The net EE of exercise would be better presented as total net EE rather than an average of the 7 exercise bouts.

●Response:

Thank you for your comments and suggestions.

i. Why were heart rate and RPE only monitored during the final 30 s of the first exercise bout and not every exercise bout?

The exercise intensity we used was of low-moderate intensity. If moderate intensity exercise is continued for 25 min, heart rate and RPE are not likely to differ significantly across the majority of this period. We wished to obtain an index of exercise stress during one bout of exercise, and we selected the last 30 sec for this purpose. However, it is possible that the HR and RPE may have differed between the bouts of exercise.

ii. Was the treadmill speed adjusted during exercise to ensure the participants were exercising at the target exercise intensity? It would be useful if additional details of the exercise sessions were included in Table 4 (e.g., exercise intensity (% peak VO2), respiratory exchange ratio, carbohydrate oxidation, fat oxidation).

We compared the %VO2 peak and actual measured VO2 values at the Lactate Threshold intensity obtained during intense PA using an aerobic capacity exercise test and a human calorimetry (HC) trial, and found no difference (p=0.064). In addition, with subjects walking, we compared the estimated VO2 values from the American College of Sports Medicine equation and the actual measured VO2 values in the chamber. We found that the actual value was higher than the estimated value (10.7±1.4 vs. 9.5±1.3 ml/min), but the difference was small. Therefore, we believe that the target exercise intensities were achieved under the exercise conditions. We have now added information regarding the responses to exercise (RQ, %VO2, HR, and RPE) in the present study in Table 3.

iii. The net EE of exercise would be better presented as total net EE rather than an average of the 7 exercise bouts.

As suggested, we have changed the values quoted to total net exercise EE (Table 3).

8.      

i: Page 5, lines 147-153: It is unclear what the ‘appetite’ VAS scale is assessing and how this differs from hunger and fullness? Hunger and fullness are ratings of perceived appetite.

ii: The time points that appetite ratings were assessed should be described in the methods and outlined on Figure 1.

iii: Why was appetite not assessed on the morning of day 2?

●Response:

In general, “hunger” and “fullness” are ascribed opposite meanings and many appetite studies [5] have asked for these to be rated at the same time. A previous review defined “appetite”, “hunger”, and “fullness” as below [6].

Appetite: a sensation related to the maintenance of eating, often involving a desire for something specific.

Hunger: a nagging, irritating feeling that signifies food deprivation to a degree that the participant wishes to eat.

Fullness: a sensation related to the degree of stomach filling.

We have added definitions regarding appetite, hunger, and fullness perception to the manuscript (Lines 169–173). In addition, we have added an explanation regarding how the VAS assessment was performed.

References

5            King, J.A.; Miyashita, M.; Wasse, L.K.; Stensel, D.J. Influence of prolonged treadmill running on appetite, energy intake and circulating concentrations of acylated ghrelin. Appetite 2010, 54, 492-498, doi:10.1016/j.appet.2010.02.002.

6               Sørensen, L.B.; Møller, P.; Flint, A.; Martens, M.; Raben, A. Effect of sensory perception of foods on appetite and food intake: a review of studies on humans. International Journal Of Obesity 2003, 27, 1152, doi:10.1038/sj.ijo.0802391.

ii. We have added the time points at which appetite was assessed to the Methods and Figure 1, as requested on line 177-182.

iii. Although we planned to assess the appetite of the participants on the morning of day 2, unfortunately we did not obtain data from two subjects at this time, and therefore we have not analyzed or shown the data obtained.

9.        Page 5, lines 154-165: Did participants empty the bladder before entering the indirect calorimeter? Were urine samples collected to analyse nitrogen content during the day? This would enable the calculation of protein oxidation and subsequently protein balance which will provide a more accurate insight into energy and substrate balance over the course of the day.

●Response:

Thank you for this suggestion. Unfortunately, we did not collect urine samples during this study, because we were mainly focusing on EB. However, we have added respiratory exchange ratio (RER) values to the manuscript instead of values for the oxidation of each substrate, because we can compare this information more readily between trials.

10.     Page 5, lines 166-167: Why was net EE only calculated for 1 h rather than the entire exercise duration (i.e., 175 min)?

●Response:

The reviewer raises a valid point. We have therefore changed the value given in the manuscript to the total net energy expenditure over the 10:00–17:30 period during which the exercise was undertaken.

11.     Page 5, lines 168-178: It would be beneficial to include 95% confidence intervals and effect sizes to supplement the P values presented in the paper. 95% confidence intervals and effect sizes provide a better insight into main effects than more traditional dichotomous hypothesis testing (Hopkins et al. Med Sci Sports Exerc 2009;41(1):3-12; Shakespeare et al. Lancet 2001;357:1349-53). Alpha values only provide an arbitrary threshold, beyond which the findings are statistically significant, which is typically dependent on sample size. However, it is possible that a non-significant result may hide an important difference that could exist but was not detected. Confidence intervals provide a range of values within which the true effect is likely to lie and provide information on the size and direction of the treatment effect compared with the control. By supplementing confidence intervals with effect sizes, we are able to identify the magnitude of the effect.

●Response:

Thank you for the valuable advice. We have added the 95% CIs and effect sizes to Tables 4 and 5 to reflect your comments.

Results

12.     Pages 5-7: The results section is slightly confusing in places and some re-wording is required to clarify the findings and confirm the trials that are being compared.

●Response:

Thank you for advice and suggestions. We greatly appreciate your well-informed comments, which we will bear in mind in the design of our future studies.

13.     Pages 6, lines 192-196: The authors could consider analysing the 24h EE in sub-sections throughout the day to examine changes in EE in different periods e.g., morning, afternoon, evening, sleep.

●Response:

We have divided the 24-h-period of each study day into sections and analyzed these separately, as per your suggestion (Table 4). We had already undertaken many of the relevant analyses, but did not show the data, because the present study focused on EB during a single day.

14.     Pages 6-7, lines 201-213: In addition to comparing energy balance between trials, details of 24-h substrate oxidation (i.e., protein, fat and carbohydrate) should be compared across trials. This is particularly relevant given the comparison of different exercise intensities which is likely to result in different patterns of substrate utilisation during exercise.

●Response:

Thank you for suggestion. As discussed above, we could not calculate the oxidation of each substrate because we did not obtain urine samples. Instead, we show respiratory exchange ratio (RER) during exercise and across the 24 h of each study day (Table 4). The RER during exercise was not different from the sedentary day, possibly because the exercise intensity used was not high enough. However, the RER over the full 24 h did show a significant difference: it was lowest during the day of high PA, reflecting greater oxidation of lipids that day.

15.     Page 6: Figure 2 should be changed to present individual participant data with a line representing the mean rather than a plunger plot which can often be misleading (Drummond & Vowler 2011 Br J Pharmacol 163: 208-210; Weissgerber et al. 2015 PLOS Biol  13: e1002128; Weissgerber et al. 2016 PLOS Biol 14: e1002484). Similar plots could also be included for substrate balance i.e., protein, carbohydrate and fat balance.

●Response:

We appreciate your suggestion. We have changed the figures to the format you have suggested (Figure 2).

Discussion

16.     Pages 7-9: The discussion section would benefit from highlighting the wider implications of the findings for energy balance particularly in the opening and concluding paragraphs.

●Response: Thank you for suggestion. We have edited the in the opening of discussion (line 266-272) and conclusion paragraphs (380-386).

17.     Page 8, lines 269-270: Include a reference to support this point.

●Response:

We have added a reference to support this point on line 331.

18.     Page 9, line 317: The mid PA trial resulted in a PA level equivalent to 1.47 x RMR not 1.3 x RMR. This should be updated.

●Response:

We have corrected the value in the text (line 380). We apologize for this error.

19.     The paper is well written and concise, but there are some errors in grammar and word choice in places which should be addressed in a revised version of the paper.

●Response:

We have submitted the manuscript to a proofreading company for it to be checked by a native English speaker, prior to the submission of a revised version.

Round 2

Reviewer 2 Report

I would like to thank the authors for revising the manuscript and addressing the comments raised. The revised manuscript has been improved but I still have some additional concerns outlined below that could be addressed in revising the manuscript.

Abstract, line 26: The units for 24 h energy intake are missing.

Abstract, lines 24-30: The presentation of the results in the abstract could be improved with the addition of P values for the presented main effects and pairwise comparisons so it is clear how energy intake, energy expenditure and energy balance are different between the conditions.

Results: I appreciate the authors efforts to address my suggestion to include 95% confidence intervals and effect sizes. However, it would be more informative to include the 95% CI of the mean absolute difference between the conditions i.e., low PA vs mid PA, low PA vs high PA and mid PA vs high PA, rather than presenting the 95% CI as a measure of dispersion within each condition separately. The 95% CI can be found in the ANOVA models if the pairwise comparison option is selected when building the models. 95% CI should be reported in this way for all tables for consistency and it will involve having 3 columns to show the 95% CI for low PA vs mid PA, low PA vs high PA and mid PA vs high PA. Furthermore, rather than calculating the ES for the main effect across the three trials, it would be more informative to calculate the ES for significant pairwise differences (i.e., low PA vs mid PA, low PA vs high PA and mid PA vs high PA) and presenting these in the text where appropriate rather than the tables. These changes will allow the reader to better understand exactly where the differences are between the conditions, rather than being restricted to the top level analysis across the three conditions through the main effect.

Results, lines 206-261: There is still some inconsistency in the presentation of the results with P values presented for some findings and not others despite the authors identifying that an effect is significant / not significant. It would help the reader to understand the differences reported if P values, supplemented with ES where appropriate, were provided consistently throughout the results.

Results, lines 221-222: It is unclear from the in-text description how the 3 conditions were different for 24 h EE and step counts. It is also unclear what the authors mean by ‘PA levels’. Specifying the main effect of trial first and if significant, presenting the post-hoc comparisons to identify the specific differences between the three conditions will help to clarify the presentation of results. This applies throughout the results section and will ensure consistency in the reporting of results.

Results, Table 4: The morning – before lunch time should be changed to 09:00-13:29. The data for 24 h EE and 24 h EB have been presented in Figure 2 so could be removed from Table 2. This will allow the table to be simplified in focusing on EE and RER during the different phases of the 24 h period.

Results, lines 221-222: It is not clear from the in-text description how the percentages of macronutrient intake were different. The inclusion of 95% CI in Table 5 which show the 95% CI of the mean differences will also facilitate better understanding of where the differences are between conditions.

Results, lines 256-257: The interaction effect was not significant for appetite so the post-hoc comparison should not be reported. The discussion section relating to this analysis (lines 347-360) should also be modified accordingly.

Discussion, lines 268-269: The interpretation of the findings in the first paragraph of the discussion is confusing and should be clarified.

Author Response

Response to Reviewer 2 Comments

Thank you for some comments and advices to improve our manuscript. Mainly we have edited the result parts as your suggestion. The point-by-point responses and the changes made to the manuscript are described below.

Abstract, line 26: The units for 24 h energy intake are missing.

●Response: Thank you for pointing this out. We have added the unit.

Abstract, lines 24-30: The presentation of the results in the abstract could be improved with the addition of P values for the presented main effects and pairwise comparisons so it is clear how energy intake, energy expenditure and energy balance are different between the conditions.

●Response: Thank you for your recommendation. We have added P values for main effect and pairwise comparisons.

Results: I appreciate the authors efforts to address my suggestion to include 95% confidence intervals and effect sizes. However, it would be more informative to include the 95% CI of the mean absolute difference between the conditions i.e., low PA vs mid PA, low PA vs high PA and mid PA vs high PA, rather than presenting the 95% CI as a measure of dispersion within each condition separately. The 95% CI can be found in the ANOVA models if the pairwise comparison option is selected when building the models. 95% CI should be reported in this way for all tables for consistency and it will involve having 3 columns to show the 95% CI for low PA vs mid PA, low PA vs high PA and mid PA vs high PA.

Furthermore, rather than calculating the ES for the main effect across the three trials, it would be more informative to calculate the ES for significant pairwise differences (i.e., low PA vs mid PA, low PA vs high PA and mid PA vs high PA) and presenting these in the text where appropriate rather than the tables. These changes will allow the reader to better understand exactly where the differences are between the conditions, rather than being restricted to the top level analysis across the three conditions through the main effect.

●Response: We greatly appreciate your advice and explanation of how to show the results. We learned many things to write the manuscript from you. Thank you so much.

We have checked and edited all the results including tables, as per your suggestion.

Results, lines 206-261: There is still some inconsistency in the presentation of the results with P values presented for some findings and not others despite the authors identifying that an effect is significant / not significant. It would help the reader to understand the differences reported if P values, supplemented with ES where appropriate, were provided consistently throughout the results.

●Response: Thank you for your comments and advice on how to improve our manuscript. As we stated above, we have added ESs.

Results, lines 221-222: It is unclear from the in-text description how the 3 conditions were different for 24 h EE and step counts. It is also unclear what the authors mean by ‘PA levels’. Specifying the main effect of trial first and if significant, presenting the post-hoc comparisons to identify the specific differences between the three conditions will help to clarify the presentation of results. This applies throughout the results section and will ensure consistency in the reporting of results.

●Response: Thank you for your advice. We have added the post hoc results of effect size and 95% CI, as per your suggestion.

Results, Table 4: The morning – before lunch time should be changed to 09:00-13:29. The data for 24 h EE and 24 h EB have been presented in Figure 2 so could be removed from Table 2. This will allow the table to be simplified in focusing on EE and RER during the different phases of the 24 h period.

●Response: Thank you for your advice. We have deleted the 24-h EE and 24-h EB in Table 4 as you suggested.

Results, lines 221-222: It is not clear from the in-text description how the percentages of macronutrient intake were different. The inclusion of 95% CI in Table 5 which show the 95% CI of the mean differences will also facilitate better understanding of where the differences are between conditions.

●Response: Thank you for your suggestions. We have added the 95% CI in Table 5, as you suggested.

Results, lines 256-257: The interaction effect was not significant for appetite so the post-hoc comparison should not be reported. The discussion section relating to this analysis (lines 347-360) should also be modified accordingly.

Response: Thank you for pointing this out. We have deleted the description of post hoc comparison for appetite perception and changed the discussion section related to this analysis. Also, we have deleted symbol marks related to the post-hoc test in Figure 3C.

Discussion, lines 268-269: The interpretation of the findings in the first paragraph of the discussion is confusing and should be clarified.

Response: Thank you for pointing this out. We have edited the first paragraph of last sentences on the discussion. We have added the opinion about effect of acute exercise on EB in lines 285–291.
